# Forest Insurance for Natural Events: An Overview by Economists

**Marielle Brunette** [1,2,*] and **Stéphane Couture** [3]

1 Université de Lorraine, Université de Strasbourg, AgroParisTech, CNRS, INRAE, BETA, 54000 Nancy, France
2 Climate Economics Chair (CEC), 75002 Paris, France
3 INRAE, MIAT, University of Toulouse, 31326 Castanet-Tolosan, France
* Correspondence: marielle.brunette@inrae.fr

**Abstract:** Forest insurance exists for more than a century in lots of countries around the world. Currently, it is put forward as a recommended tool to finance resilience and adaptation towards climate change. However, little synthetic knowledge exists on forest insurance, although this seems to be a prerequisite for using insurance as an adaptation tool. This article aims at providing an overview of the current economics literature on the topic of forest insurance. More precisely, the objectives of this study are to carry out a review of the literature on this topic, to produce a bibliometric overview of knowledge on this issue, and thus to highlight scientific fronts. For that purpose, we propose a literature review. We collected 38 articles published in English between 1928 and 2021. We provide the following bibliometric information: journals, evolution over time of the publications, authors and co-citations network and analysis of the keywords. We also propose to synthesize the methods used, the various issues of interest, the risks considered and the countries where the studies were conducted. We show that an article on forest insurance has a high probability of being recent (after 2000) and of being published in the journal Forest Policy and Economics. In addition, it is highly probable that it will identify some determinants of insurance demand and that it will deal with fire risk in the U.S. or storm risk in Europe. Noting a small scientific community and a low number of publications, we identified seven fronts of science related to methods and data, new risks and uncertainties, public policies and forest insurance, and openness and the international dimension.

**Keywords:** risk; forest; insurance; literature review

## 1. Introduction

Natural hazards are the main threat to forests worldwide. It has been estimated that over the period of 2002–2013, 67 million hectares of forest were annually burned worldwide, 85 million hectares were affected by insects, 38 million by severe weather conditions and 12.5 million by disease [1]. At the European scale, [2] indicated that over the period of 1950–2000, an annual average of 35 million $m^3$ of wood was damaged by disturbances in Europe. Storms were responsible for 53% of the total damage, fire for 16%, snow for 3% and biotic factors for 16%. In addition, [2] shows that disturbances increased over the period. This increase continues in the first decade of the 21st century [3]. More importantly, damage from these disturbances is likely to increase even more in the coming decades [3]. Climate change was identified as a main driver behind this increase [4], and it impacts both the frequency and intensity of disturbances [5].

Insurance for these natural disturbances is available in many countries. Indeed, forest owners can transfer risks to insurers through an insurance contract. It is generally fire and storm damages that can be insured, like in France and Germany, for example, or other risks, such as insect damage in Finland or carbon loss in New Zealand [6]. However, large differences can be observed among countries in terms of insurance adoption. The penetration rate of forest insurance is the highest on the market in northern countries,

with 95% of the private forest area insured in Sweden, and around 40% in Finland and Norway [7]. In other countries, the situation is very different. In France, less than 4% of the private forest area is insured, and a similar situation exists in Spain and Germany [8].

In the context of increasing risks due to climate change, insurance is recommended as a vehicle to finance climate resilience and adaptation by several organizations, such as the Global Agenda Council on Climate Change (2014), OECD (2015), UNFCCC (Article 4.8) and the Kyoto Protocol (Article 3.14). This recommendation raises many questions concerning the reasons for the differences in terms of insurance adoption between countries, the way the current insurance scheme should be adapted to face the new challenge of climate change, the role of the public authorities, etc. However, before addressing these questions, a first step dedicated to synthesis is necessary. This article aims at providing an overview of the current economics literature on the topic of forest insurance. More precisely, the objectives of this article are the following: to carry out a review of the economic literature on forest insurance, to produce a bibliometric overview of knowledge on this issue and to highlight scientific fronts. We will have a look at various information like the methodologies prioritized in the literature, the main issues addressed, the risks that are of interest, and the countries where the case studies take place. For that purpose, we propose a literature review. We carried out a systematic search on Google Scholar and identified 38 relevant articles dealing with forest insurance that we synthesized and classified in this paper. We noted a small scientific community and a low number of publications. We identified seven fronts of science related to methods and data, new risks and uncertainties, public policies and forest insurance, and openness and the international dimension.

This study is organized as follows: Section 2 provides the materials and methods that have been used. Section 3 presents the results that are then discussed in Section 4. Finally, Section 5 concludes.

## 2. Materials and Methods

### 2.1. The Methodology for the Systematic Search

In order to provide an overview of the literature on forest insurance, we carried out a literature review. More precisely, we implemented a systematic search on Google Scholar. We opted for the Google Scholar research tool because, first, our bibliometric study focuses on a very specific topic on which, as specialists, we had a preconception of a very limited literature and, second, we already had a precise knowledge of the existing literature on the subject and we had strict selection criteria more in line with such a tool. We realized the search with the two words "Insurance AND Forest". Such systematic search has already been conducted in economics to tackle problems that are close to ours, see for example [9–12] but also in other fields such as neuromarketing [13] and healthcare [14]. The search was terminated in June 2022. We collected 38 articles containing the words "Insurance" and "Forest" (title, abstract, or text) with the first article published in 1928 and the last in 2021 (see Appendix A). Table 1 shows the process of selecting articles for this study. Finally, we only selected articles published in peer-reviewed journals in English.

**Table 1.** Flow chart to identify articles for the bibliometrics study.

| Stage | Selection | Exclusion |
| --- | --- | --- |
| Identification | Records identified through Google Scholar searching | Query: «Insurance» and «forest» |
| Screening | Record screened based on title, keywords and year | Excluded due to non-compliance with the area |
| | Record screened based on type of publication | Excluded not only peer-reviewed articles |
| Eligibility | Full articles published in English | Excluded due to non English language |
| Selection | Articles included in the bibliometric study | |

As a first step, a bibliometric analysis is carried out on this database.

*2.2. Some Bibliometric Information on the Database*

2.2.1. Temporal evolution

We can easily observe in Figure 1 that the literature is recent. Indeed, although the seminal papers were published during the decade 1920–1930, most of the literature on the topic appears after 2000 and even after 2010. The reason is probably that in the context of climate change, insurance tools are promoted and recommended by public institutions, encouraging research on this topic.

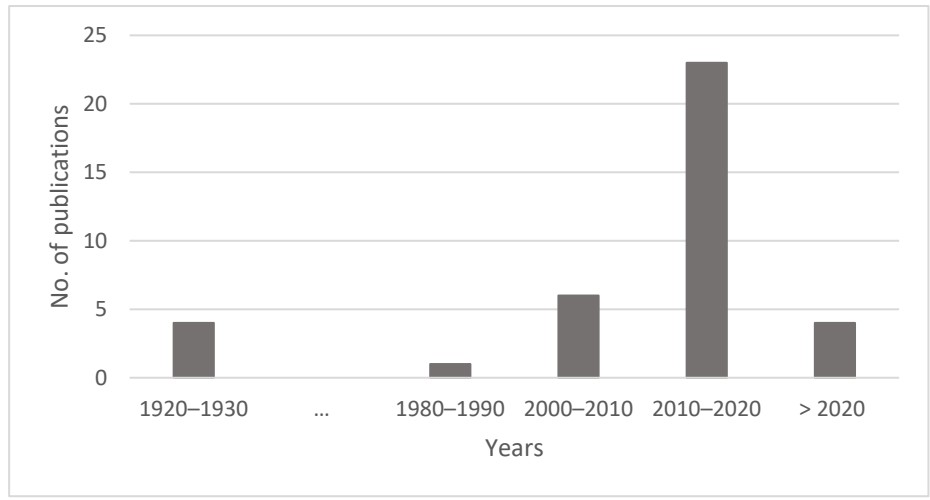

**Figure 1.** Temporal evolution of the number of publications on forest insurance within our sample of 38 articles.

2.2.2. Journals

Table 2 presents the three journals with the higher number of publications. Table 2 reveals that approximately 40% of the articles were published in the following three journals: *Forest Policy and Economics*, *Journal of Forestry* and *Ecological Economics*. In addition, the four papers published in the *Journal of Forestry* are those that were published during the decade 1920-1930. All the other articles were published in a different journal as follows: forestry-oriented (*Forest Fire Prevention and Control, Annals of Forest Science*, etc.), with an agricultural thematic (*Canadian Journal of Agricultural Economics, Journal of Agricultural and Applied Economics,* etc.), more general dealing with environment (*Environmental and Resource Economics, Journal of Environmental Management*, etc.) or generalist journals in economics (*Theory and Decision, The Geneva Papers on Risk and Insurance—Issues and Practices*, etc.).

**Table 2.** The journals with the higher number of publications.

| Journal | No. of Articles |
|---|---|
| Forest Policy and Economics | 8 |
| Journal of Forestry | 4 |
| Ecological Economics | 3 |

2.2.3. Authors

The 38 articles are written by 80 authors. In Figure 2 we propose to take the name of the authors and look at their number of occurrences (size of the circle) and the links between them (the size and number of links). The size of the circle is proportional to the number of occurrences of the author's name. The names are linked together if they appear at least one time in the same publication, and the thickness of the link between names is proportional to their number of co-occurrences. The colors allow us to identify clusters of authors. We created Figure 2 with VOSviewer, an open-access software tool developed to construct and visualize bibliometric networks [15]. We used this software to easily and

fashionably visualize the links (in terms of publications) between the authors (Figure 2) and the keywords (Figure 3).

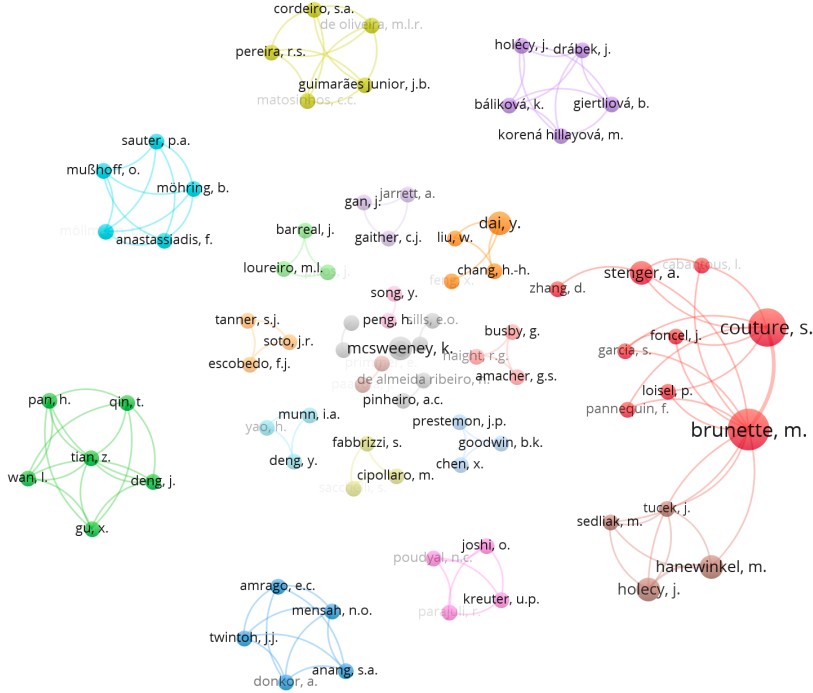

**Figure 2.** The networks of the authors and co-citations, created with VOSViewer.

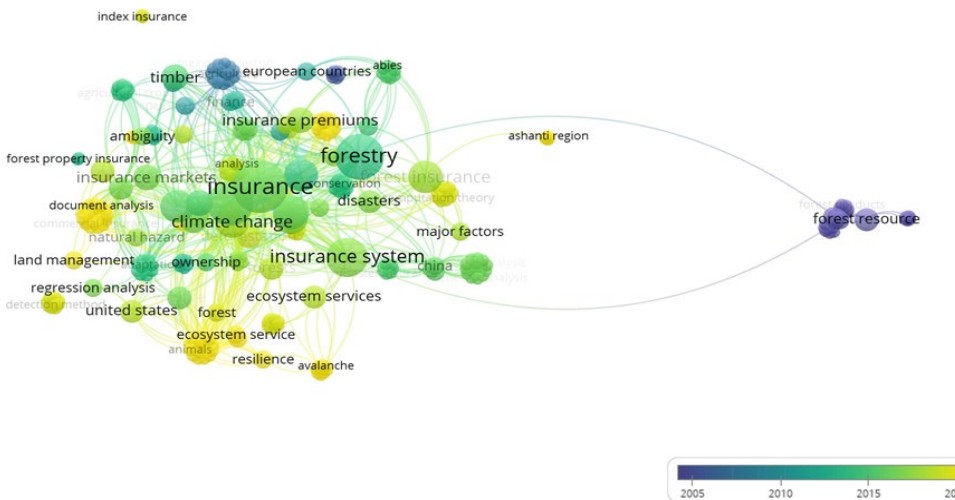

**Figure 3.** Index keyword co-occurrence network for 30 articles, created with VOSViewer.

We can observe that the clusters are numerous and are very few connected. The co-citations networks are very small in general, with the higher cluster gathering 13 authors in red/brown colors.

### 2.2.4. Keywords

In Figure 3, we propose to have a similar analysis. We take the keywords stated by the authors on the article's title page and look at their number of occurrences (the size of the circle) and the links between them (the size and number of links). In addition, the color represents the evolution in time, from blue in 2005 to yellow in 2020, through green from 2010 to 2015. Thirty articles are considered for this graph; the eight others have no keywords (the eight papers published before 2004).

Not surprisingly, the two words with the highest number of occurrences are "insurance" and "forestry". We also find "insurance system" and "climate change". Topics related to forest resources and European countries are in blue (the older, in 2005), whereas the occurrences in yellow (more recent, in 2020) deal with ecosystem services and resilience.

## 3. Results

From these first elements of the bibliometric analysis, it is possible to draw up recommendations and a bibliometric overview of knowledge on this issue, in particular on the methods currently used, the main risks studied, the subjects targeted and finally, the countries where forest insurance is a key issue.

### 3.1. The Prioritized Methodologies: Theoretical Model or Empirical Analysis

A look at the 38 articles reveals that one of the following two types of methodologies is mobilized to tackle the topic of forest insurance in economics: theoretical models or empirical analysis. Indeed, real data on forest insurance are not available, so that researchers working on this topic use either theoretical models to address their research questions or empirical methods to collect their own data. Table 3 summarizes the methods, objective and the references for each methodology.

**Table 3.** Summary of the methodologies.

| Theoretical Model | | | Empirical Analysis | | |
|---|---|---|---|---|---|
| **Methods** | **Objective** | **Articles** | **Methods** | **Objective** | **Articles** |
| Actuarial model | Provide tools to compute forest insurance premium | [8,16,17] | Experimental economics | Estimate forest owners' willingness-to-pay for forest insurance | [19,30–35] |
| Insurance economics model | Model the forest owner's behavior in terms of insurance | [18–21] | Surveys | Gather data on the determinants of the forest insurance demand | [36–43] |
| Forest economics and cost-benefit analysis | Introduce insurance decisions into classical forest economics models | [22–25] | Descriptive analyses/Reports/Reviews | Description of forest insurance schemes in certain countries, or comparison between countries | [7,44–50] |
| Risk models and spatialization | Model forest risks and introduce spatial dimension | [25–27] | | | |
| Others | Multi-agent simulation models | [28] | | | |
| | Stackelberg game models | [29] | | | |

### 3.2. The Main Issues Addressed in the Literature

Recurring topics appeared in the articles. We present some of them in this section.

Some articles deal with the impact of public assistance on forest owners' insurance decisions. They focus on various types of assistance such as subsidy of the insurance premium, conditional (to the subscription of an insurance contract) and unconditional assistance [18,19,28–30,33,36]. The results depend on the type of public assistance considered. For example, Ref. [33] shows that public programs involving unconditional support after a disaster reduce the willingness-to-pay for insurance, whereas subsidized insurance premiums have a positive impact. More generally, these articles show that public assistance, conditional or not, discourages owners from adopting insurance. They extend the "charity hazard" problem to forestry. Indeed, Ref. [51], dealing with flooding issues, defined the "charity hazard" as the tendency not to insure as a result of the reliance on expected financial assistance from federal relief programs.

Another recurring topic is natural insurance [34,35,40,42,43,48]. Indeed, in the forestry sector, insurance can be of two types. Forest owners can take out an insurance contract with

an insurer to cover themselves against damage attributable to natural hazards (financial insurance) and at the same time, the forest provides protection against natural hazards to exposed populations (natural insurance). Consequently, forests may be covered against natural events by insurance contracts, but forests may also be the insurer in the sense that forests prevent some natural events from occurring (protection role), i.e., gravitational risks (avalanches, rockfalls, landslides). Financial insurance was historically the first topic addressed in the literature. However, recently, some articles have begun to deal with natural insurance. These articles are based on the more general idea that ecosystems can act as a buffer against sudden adverse events and incremental deterioration and losses, and thereby provide insurance value [52]. The articles examine case studies in Switzerland [35], Honduras [42,43], the southeastern U.S. [34] and Brazil [40] and attempt to determine the insurance value of some ecosystems. Ref. [48] adopts a more general approach and propose to deal with the governance of ecosystem service provision.

Indirectly, these topics reveal the potential explanations for the heterogeneity of forest insurance behavior between countries. Indeed, the way public assistance is disposed of is different from one country to another, potentially explaining the difference. The insurance value of forest ecosystems is not considered, explaining the low subscription rate of owners in countries where this natural insurance is high.

Other articles directly aim to identify determinants of insurance demand in forests [30,31,33,37,39,41]. Using empirical approaches, they question the forest owner's motivation. For example, Ref. [39] shows that prescribed burn practitioners in the U.S. are more likely to insure if they are landowners themselves or have a written prescribed burn plan. The age of respondents and the level of importance they place on compliance with environmental laws also have a significant positive effect, whereas the land management objective to control invasive plants has a negative impact. Ref. [31], in Mississippi (U.S.), showed that landowners were willing to pay a higher premium rate if their primary goal for timberland ownership was timber revenue, if they were risk averse and had substantial income, if they had previously experienced a loss and if they were concerned about risks to standing timber. We can observe that the results are not always unanimous. For example, having recently suffered from a natural event has a significant and positive effect on insurance demand for [30,31], whereas it has a negative one for [37]. In the same way, the higher the forest owner's income is, the higher the probability to insure will be for [30,31], but not for [37], who observed no effect of income level.

To conclude, it should be noted that among the other topics analyzed, we have forest insurance in the context of climate change [20], carbon insurance [29] and the spatial dimension of forest insurance [25].

*3.3. The Risks Considered: From Wildfire in U.S. to Storm in Europe*

We observe that the literature mainly focuses on insurance against wildfire [17,19,23,26,30,33,37,39,45–47,49,50]. This risk is at the origin of this literature on forest insurance since the seminal papers [45,47,49,50] focus on wildfire insurance in the U.S. and Japan. However, storms are also largely studied, especially in Europe [18,22,33,53], and hurricanes in Honduras [43]. Pine beetles are rarely studied [27]. Finally, some articles deal with multiple risks at the same time, either questioning the willingness-to-pay for several risks [31] or dealing with their cumulative aspect [8]. Some authors have specifically considered the following most damageable events for European forests: fire and storm [25,33]. On this last point, Ref. [33] obtained the following interesting result: German forest owners' willingness-to-pay is higher for fire than for storm insurance.

*3.4. The Countries Where Forest Insurance Is an Issue: Europe . . . but Not Only*

We observed that many articles deal with the United States [27,31,32,34,37,39,45,47,49]. They were the first to address the question of forest insurance, in particular, fire risk. The first articles in the database are [45] and [47]. However, the literature also covers other countries such as Spain [23], Italy [25], China [21,29,36], Sweden [44], Slovakia [8,46],

Germany [16,33], Ghana [38], Brazil [24,40], Portugal [17], Honduras [42,43], Japan [50], France [19,22,30], Switzerland [35,53] and New Zealand [6]. At the continental level, 13 articles related to Europe, 4 to Asia, 9 to North America, 4 to South America, 1 to Oceania and 1 to Africa. Note that the sum is 32 since some articles are very general and not focused on one country in particular. They are mainly theoretical papers with no case studies. For example, [18] proposed a theoretical model of insurance demand that tested the impact of various types of public assistance. Another example is [26], which examines the fuel treatment decision between adjacent landowners where spatial externalities are present.

## 4. Discussion

This synthesis and the associated detailed analysis allow us to identify potential gaps in the literature and avenues for future research. On the basis of this analysis, it appears that the community of researchers working on the issue of forest insurance is small and that the number of publications is limited in the face of a complex and major problem. There is a need for methodological and empirical advances to meet the challenges of this issue in an uncertain context of global change. More precisely, we identify seven "fronts of science", i.e., increasing topics in economics to which more attention should be paid in the near future, dealing with the following four main thematics: methods and data, new risks and uncertainties, public policies and forest insurance, and openness and international dimension.

### 4.1. Methods and Data

The problem of insurance choices in forests cannot be studied alone because other actions or decisions taken by the forest owner will impact the insured resource. It is mainly decisions related to silvicultural practices that will modify the value of the insured property or the possible losses in case of a claim. Some silvicultural actions can also be considered as direct coverage measures against some risks, particularly climatic or biological. For example, reducing rotation length or stand density or planting more well-adapted species are considered adaptation strategies. Therefore, it is essential to link forest management approaches of a dynamic nature with the static nature of the insurance decision-making process. Various problems of different scales may then arise. For example, in a dynamic framework, insurance may be perceived as a risky investment where the occurrence of a random event may be favorably perceived by certain decision-makers. Moreover, behaviors related to silvicultural forest management practices can be seen as internal hedging and self-insurance tools, which can be considered either complementary to or substitutable for private insurance. Thus, such decisions will be important for the insurance choice. The first front of science is then to consider more complex models to study the insurance/silvicultural practices interaction in a dynamic setting. Even if such models prove to be a tool to be explored, they will require observational data to calibrate them. At present, there is a severe lack of usable data on observed and future behavior in terms of the insurance. Therefore, it is very difficult to conduct empirical and econometric analyses of such behavior and to identify the key factors that explain such decisions, which are indispensable for complex decision-making models. Judging from the few empirical studies found in the literature, it appears that the identified determinants of insurance demand are related to the socio-economic characteristics of forest owners, such as age and income level, and to the characteristics of the forest (land management objective, recent loss, ownership). It would also be very useful to highlight the importance of behavioral factors that generally explain an insurance decision, such as, for example, attitudes towards risk and uncertainty, perceptions of risk and climate change, etc., the various internal psychological and cognitive components that are not directly observable but are fundamental to understand the decision-making process in a risky environment. It would be possible to use experimental techniques to recover data and explore vague or ambiguous areas of this problem, but the use of such approaches is currently limited

because of various biases related to the tool. A second front of science linked to the establishment of a database that would make it possible to make robust empirical studies is emerging.

### 4.2. New Risks and Uncertainties

Previous works on forest insurance mainly focus on "classical risks" in forests, such as storms and fires, based on traditional forest insurance schemes, i.e., the forest owner pays a premium in exchange for receiving an indemnity in the event of natural hazard occurrence. However, under the influence of climate change, other risks have become relevant in forestry, such as drought, pests and pathogens, and to consider these "new risks", it may be helpful to enlarge the possibilities in terms of insurance schemes. For example, in agriculture, the following interesting alternative to this type of traditional insurance scheme exists: the parametric or indexed insurance. The idea is that the indemnity is distributed to the farmer since the threshold level of an index is exceeded. Ref. [54] studied this indexed insurance and proved the feasibility of such a parametric insurance to insure drought risk in a French forest. They considered different types of indexes, ranging from simple ones that rely on rainfall indexes to complex ones that rely on the functional modeling of forest water stress. We can easily imagine covering other risks with such an insurance, with indexes such as the speed of wind for storms or the surface area burned for fire. Another interesting research area could also deal with the definition of relevant indexes for these "new risks", for example, for pests and pathogens. Aerial observation of the coloration of the canopy would then be helpful. Thinking about new insurance mechanisms for "new risks" is a third front in science.

Currently, the decision of a forest owner to insure is made in a certain environment and in the face of global changes. Indeed, the resource is confronted simultaneously or independently with multiple risks. Indeed, a drought year may favor forest fire occurrence, a storm occurrence may favor insect invasion, etc. The risks may be simultaneous or sequential [54], but they are correlated, which reinforces the difficulty of their insurability. Therefore, it is necessary to reason within a framework of spatially and temporally correlated multiple risks and to thus propose avenues of research toward new tools of coverage or adapted insurance. We should no longer think in terms of a single risk but in terms of multiple risks. In such a framework, the importance of preferences and the background risk effect will be highlighted, even for independent risks. The literature already contains articles that indirectly deal with these multi-risk aspects [8,12,31] and, more recently, with the spatial dimension [25], but the trend should be stressed. Consequently, we consider the need to take the multi-risk dimension of the problems more systematically into account both from a spatial and temporal point of view as a fourth scientific front.

Moreover, the global changes that forest owners face, such as climate change or uncertain geo-political or socio-economic situations, are by nature uncertain. Therefore, an approach in the context of uncertainties and not of risks is fundamental. Indeed, most of the existing literature considers that negative events are well-known both in terms of frequency and intensity, whereas climate change impacts these characteristics. In this changing context, it is of utmost importance to consider that the characteristics of these events are imprecise. This difference between risk and uncertainty refers to the typology of [55]. Furthermore, this uncertainty raises the question of insurability. This results in a fifth front of science where an approach that considers uncertainty should be taken.

### 4.3. Public Policies and Forest Insurance

In such uncertain situations and in the face of new emerging risks where little information may exist, making their insurance complicated, public intervention may be fundamental. On that point, the literature has already shown that certain types of public assistance seem to discourage forest owners from adopting insurance contracts. Consequently, dealing with new ways of public intervention may be relevant. We can imagine subsidizing the insurance premium, as is the case in Europe for agricultural insurance,

in the context of the Common Agricultural Policy. We can also imagine integrating the adaptation efforts of forest owners towards climate change in the insurance premium computation [20]. Indeed, implementing adaptation strategies such as the reduction of density, the planting of more well-adapted tree species or the reduction of rotation age would make it possible to reduce risk exposure. Consequently, these practices (and their associated costs) could be considered when computing the insurance premium. Payment for Environmental Services (PES) may be another potential vector for public assistance. Indeed, carbon sequestration is at the heart of the international debate on climate change, and forests clearly represent a carbon sink. Such a tool may contribute to the adoption of insurance by forest owners. For example, Ref. [48] propose to use them to secure the role of natural forest insurance. Thus, the weight of public policies on insurance decisions may be key and merits further study. In any case, in this context, such public policies will condition insurance adoption. A sixth front of science dealing with public policy analysis in the context of forest insurance is emerging.

*4.4. Openness and International Dimension*

Finally, studies on forest insurance are generally of a punctual nature, both spatially and temporally. Most of them are based on very specific case studies (one region or country, one tree species, one risk, etc.) questioning the generalization of the results. However, the current context is at a different level; it is now necessary to think in terms of a broader spatial dimension or even international coverage. For example, Refs. [8,16] suggest the implementation of a European forest insurance scheme to increase pooling and to make it possible to propose lower insurance premiums. Studies on a large spatial scale will be crucial to address the issue of forest insurance. This is the seventh front of science.

Other ambitions and problems also arise for forest insurance. For example, we can mention the ecosystem services provided by the forest (biodiversity, recreation, carbon sequestration, etc.), which are also impacted by the occurrence of natural events. A legitimate question is then the following: what type of insurance mechanism can be proposed to cover the loss of ecosystem services? For example, in the context of climate change, where countries are engaged in international negotiations to reduce CO2 emissions, the forest sink is an important tool. In some countries where the private forest area is important, it may be helpful to financially compensate private forest owners for their efforts to sequestrate carbon as part of their forest management. This is consistent with the idea of [6,29], who propose to secure forestry carbon sink through insurance. According to [29], "insurance can ensure the rapid recovery of forestry production and operation after the disaster, reduce the risk of forestry investment and financing, and promote the sustainable operation of forestry". For [6], the idea is to increase the sum insured per hectare to cover the carbon value as well as the tree crop value. These are also scientific challenges to be met.

## 5. Conclusions

This article gives an overview of the existing literature on forest insurance. We provide some bibliometric information about the journals in which the articles were published and the evolution over time of the number of published articles. We also propose to synthesize several types of information, such as the methods used to address the research question, the various issues addressed, the risks considered and the countries interested in the topic of forest insurance. We can sum up our results as follows: an article on forest insurance has a high probability of being recent (after 2000) and of being published in the journal Forest Policy and Economics. In addition, it is highly probable that it will identify some determinants of insurance demand and that it will deal with fire risk in the U.S. or storm risk in Europe.

Some limitations of the current study may be underlined. For example, we restrict our research to articles published in peer-reviewed journals in English. This means that all other documents are excluded, whereas we know that some of them may be interesting and important for the research community, such as [6,54] for example. Another limitation

is about the specific keywords we used, "Insurance AND Forest". We really focus on the two words relevant to the article. However, other words may be tested, such as "Insurance AND timber", for example. Finally, the research was conducted on Google Scholar, whereas other search engines may be mobilized.

In the synthesis of the existing literature being performed, lots of questions stay unexplored or without a clear answer. Future research should thus focus on understanding the differences in terms of insurance adoption between countries or the way the current insurance scheme should be adapted to face the new challenge of climate change. This article is the first step of a more general reflection on forest insurance that will pursue.

**Author Contributions:** M.B. realized the synthesis of the literature review and S.C. writes the discussion. The article was written by both authors. The article was revised by both authors. All authors have read and agreed to the published version of the manuscript.

**Funding:** This project received funding from the European Union's Horizon Europe research and innovation program under grant agreement No. 101059498.

**Data Availability Statement:** Not applicable.

**Acknowledgments:** We are very grateful to Félix Bastit for the help provided with VOSViewer.

**Conflicts of Interest:** The authors declare no conflict of interest.

## Appendix A

- Angström, A. Forest insurance in Sweden. *For. Fire Prev. Cont.* **1982**, *7*, 223–227. https://doi.org/10.1007/978-94-017-1574-4_26.
- Barreal, J.; Loureiro, M.; Picos, J. On insurance as a tool for securing forest restoration after wildfires. *Forest Policy Econ.* **2014**, *42*, 15–23. https://doi.org/10.1016/j.forpol.2014.02.001.
- Brown, W.R. Forest fire actuary. *J. Forest* **1928**, *26*, 88–90. https://doi.org/10.1093/jof/26.1.88.
- Brunette, M.; Couture, S. Public compensation for windstorm damage reduces incentives for risk management investments. *Forest Policy Econ.* **2008**, *10*, 491–499. https://doi.org/10.1016/j.forpol.2008.05.001.
- Brunette. M.; Cabantous, L.; Couture, S.; Stenger, A. The impact of governmental assistance on insurance demand under ambiguity: A theoretical model and an experimental test. *Theor. Decis* **2013**, *75*, 153–174. https://doi.org/10.1007/s11238-012-9321-8.
- Brunette. M.; Holecy, J.; Sedliak, M.; Tucek, J.; Hanewinkel, M. An actuarial model of forest insurance against multiple natural hazards in fir (abies alba mill.) stands in Slovakia. *Forest Policy Econ.* **2015**, *55*, 46–57. https://doi.org/10.1016/j.forpol.2015.03.001.
- Brunette, M.; Couture, S.; Pannequin, F. Is forest insurance a relevant vector to induce adaptation efforts to climate change? *Ann. For. Sci* **2017**, *74*, 41. https://doi.org/10.1007/s13595-017-0639-9.
- Brunette, M.; Couture, S.; Foncel, J.; Garcia, S. The decision to insure against forest fire risk: An econometric analysis combining hypothetical and real data. *Geneva Pap. R IISS P* **2020**, *45*, 111–133. https://doi.org/10.1057/s41288-019-00146-6.
- Busby, G.; Amacher, G.S.; Haight, R.G. The social costs of homeowner decisions in fire-prone communities: Information, insurance, and amenities. *Ecol. Econ.* **2013**, *92*, 104–113. https://doi.org/10.1016/j.ecolecon.2013.02.019.
- Chen, X.; Goodwin, B.K.; Prestemon, J.P. How to fight against southern pine beetle epidemics: An insurance approach. *Can. J. Agr. Econ.* **2019**, *67*, 193–213. https://doi.org/10.1111/cjag.12186.
- Dai, Y.; Chang, H.H.; Liu, W. Do forest producers benefit from the forest disaster insurance program? Empirical evidence in Fujian Province of China. *Forest Policy Econ.* **2015**, *50*, 127–133. https://doi.org/10.1016/j.forpol.2014.06.001.

- Deng, Y.; Munn, I.A.; Coble, K.H. Willingness-to-pay for potential standing timber insurance. *J. Agr. App. Econ.* **2015**, *47*, 510–538. https://doi.org/10.1016/j.forpol.2014.06.001.
- Deng, Y.; Munn, I.A.; Yiao, H. Attributes-based conjoint analysis of landowner preferences for standing timber insurance. *Risk Manag. Ins. Rev.* **2021**, *24*, 421–444. https://doi.org/10.1111/rmir.12196.
- Feng, X.; Dai, Y. An innovative type of forest insurance in China based on the robust approach. *Forest Policy Econ.* **2019**, *104*, 23–32. https://doi.org/10.1016/j.forpol.2019.03.012.
- Gan, J.; Jarrett, A.; Johnson Gaither, C. Wildfire risk adaptation: propensity of forestland owners to purchase wildfire insurance in the southern United States. *Can. J. For. Res.* **2014**, *44*, 1376–1382. https://doi.org/10.1139/cjfr-2014-0301.
- Hillayová, M.K.; Báliková, K.; Giertliová, B.; Drábek, J.; Holécy, J. Possibilities of forest property insurance against the risk of fire in Slovakia. *J. For. Sci.* **2021**, *67*, 204–211. https://doi.org/10.17221/199/2020-JFS.
- Holecy, J.; Hanewinkel, M. A forest management risk insurance model and its application to coniferous stands in southwest Germany. *Forest Policy Econ.* **2006**, *8*, 161–174. https://doi.org/10.1016/j.forpol.2004.05.009.
- Holthausen, N.; Baur. On the demand for an insurance against storm damage in Swiss forests. *J. For. Suisse* **2004**.
- Kaul, J. Report of Committee on Forest Fire Insurance of the Commercial Forestry Conference. *J. Forest* **1928**, *26*, 76–84. https://doi.org/10.1093/jof/26.1.76.
- Loisel, P.; Brunette, M.; Couture, S. Insurance and forest rotation decisions under storm risk. *Environ. Res. Econ.* **2020**, *76*, 347–367. https://doi.org/10.1007/s10640-020-00429-w.
- Ma, N.; Li, C.; Zuo, Y. Research on forest insurance policy simulation in China. *Forest Econ. Rev.* **2019**, *1*, 82–95. https://doi.org/10.1108/FER-03-2019-0004.
- Mc Sweeney, K. Natural insurance, forest access, and compounded misfortune: Forest resources in smallholder coping strategies before and after Hurricane Mitch, northeastern Honduras. *World Dev.* **2005**, *33*, 1453–1471. https://doi.org/10.1016/j.worlddev.2004.10.008.
- Mc Sweeney, K. Forest product sale as natural insurance: the effects of household characteristics and the nature of shock in eastern Honduras. *Soc. Natur. Resour.* **2004**, *17*, 39–56. https://doi.org/10.1080/08941920490247245.
- Mensah, N.O.; Twintoh, J.J.; Amrago, E.C.; Donkor, A.; Anang, S.A. Forestry insurance preference among tree growers in the Ashanti Region of Ghana: a tobit and multinomial regression approach. *Manag. Financ.* **2021**, *47*, 1194–1212. https://doi.org/10.1108/MF-10-2020-0535.
- Paavola, J.; Primmer, E. Governing the provision of insurance value from ecosystems. *Ecol. Econ.* **2019**, *164*, 106346. https://doi.org/10.1016/j.ecolecon.2019.06.001.
- Parajuli, R.; Joshi, O.; Poudyal, N.C.; Kreuter, U.P. To Insure or not to Insure? Factors Affecting Acquisition of Prescribed Burning Insurance Coverage. *Rangeland Ecol. Manag.* **2019**, *72*, 968–975. https://doi.org/10.1016/j.rama.2019.07.007.
- Pattanayak, S.K.; Sills, E.O. Do tropical forests provide natural insurance? The microeconomics of non-timber forest product collection in the Brazilian Amazon. *Land Econ.* **2001**, *77*, 595–612. https://doi.org/10.2307/3146943.
- Pereira, R.S.; Araujo Cordeiro, S.; Leles Romarco de Oliveira, M.; Matosinhos, C.C.; Benedito Guimarães Junior, J. Cost of forest insurance in the economic viability of eucalyptus plants. *Revista Árvore* **2018**, *42*, e420302. https://doi.org/10.1590/1806-90882018000300002.
- Pinheiro, A.; Ribeiro, N. Forest property insurance: an application to Portuguese woodlands. *Intern. J. Sustain. Soc.* **2013**, *5*, 284–295. https://doi.org/10.1504/IJSSOC.2013.054716.

- Qin, T.; Gu, X.; Tian, Z.; Pan, H.; Deng, J.; Wan, L. An empirical analysis of the factors influencing farmer demand for forest insurance: Based on surveys from Lin'an County in Zhejiang Province of China. *J. Forest Econ.* **2016**, *24*, 37–51. https://doi.org/10.1016/j.jfe.2016.04.001.
- Sacchelli, S.; Cipollaro, M.; Fabbrizzi, S. A Gis-based model for multiscale forest insurance analysis: The Italian case study. *Forest Policy Econ.* **2018**, *92*, 106–118. https://doi.org/10.1016/j.forpol.2018.04.011.
- Sauter, P.; Mollmann, T.B.; Anastassiadis, F.; Musshoff, O.; Mohring, B. To insure or not to insure? Analysis of foresters' willingness-to-pay for fire and storm insurance. *Forest Policy Econ.* **2016**, *73*, 78–89. https://doi.org/10.1016/j.forpol.2016.08.005.
- Shepard, H.B. Forest Fire Insurance in the Pacific Coast States. *J. Forest* **1935**, *33*, 111–116. https://doi.org/10.1093/jof/33.2.111.
- Song, Y.; Peng, H. Strategies of Forestry Carbon Sink under Forest Insurance and Subsidies. *Sustainability* **2019**, *11*, 4607. https://doi.org/10.3390/su11174607.
- Tanner, S.J.; Escobedo, F.J.; Soto, S.R. Recognizing the insurance value of resilience: Evidence from a forest restoration policy in the southeastern US. *J. Environ. Manag.* **2021**, *289*, 112442. https://doi.org/10.1016/j.jenvman.2021.112442.
- Unterberger, C.; Olschewski, R. Determining the insurance value of ecosystems: A discrete choice study on natural hazard protection by forests. *Ecol Econ.* **2021**, *180*, 106866. https://doi.org/10.1016/j.ecolecon.2020.106866.
- Yatagai, M. History and Present Status of Forest Fire Insurance in Japan. *J. Forest* **1933**, *31*, 79–84. https://doi.org/10.1093/jof/31.1.79.
- Zhang, D.; Stenger, A. Timber insurance: Perspectives from a legal case and a preliminary review of practices throughout the world. *New Zeal. J. For. Sci* **2014**, *44*, s9. https://doi.org/10.1186/1179-5395-44-S1-S9.

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
