# Peer review of "Forest Insurance for Natural Events: An Overview by Economists"

_forests, doi:10.3390/f14020289_

Round 1
Reviewer 1 Report
REVIEW COMMENTS
After the review process, the reviewer would like to give some critical thinking and idea to help authors get their job done efficiently.
I have only a few concerns about the paper and some suggestions that maybe the authors could consider:
1. To begin with, there are some typos and grammar mistakes. Some long sentences could make readers confused.
2. In the 'Introduction' section, the proposed research gap and the stated objectives do not meet the criteria of proper synergy. Please make the research gap and the research objectives consistent with each other.
3. In the 'Introduction' section, I would like to suggest the authors add a paragraph at the end of the 'introduction' to write the organization of the paper to ease readers to catch the overview of the article's content. You may check an example in this ref. 'current trends in the application of eeg in neuromarketing: a bibliometric analysis'.
4. I think the 'Material and Methods' section can be improved by adding a new bibliometric analysis of articles from different fields to highlight the importance of bibliometric analysis, which has recently become widely used, which can be beneficial for this issue.
5. It might be appropriate for the authors to explain why they had chosen the period of extracting data between 1928 and 2021.
6. Compared with the period, I think the number of extracted articles (i.e., 38) is quite small. So, the authors should clarify that.
7. Why have you chosen these specific keywords, i.e., "Insurance AND Forest". I think it's broad keywords.
8. Figure 1 needs more clarity for public readers. For example, X-axis and Y-axis refer to what?
9. The authors did not refer to the type of extracted documents that have been analyzed (e.g., article, review paper, ..., etc.). In addition, the authors should refer to the language of documents that have been analyzed in this study. So, I suggest a ref. 'neuromarketing research in the last five years- a bibliometric analysis', which can benefit and improve the issue.
10. Could the authors clarify in the article what type of documents has been excluded from the Google Scholar database for this study?
11. The development of search criteria does not justify why the decisions are made. So, the authors should clarify why they have chosen the Google Scholar database for extraction papers and not in Scopus or WoS database.
12. Could the authors explain why they have used VOSviewer software rather than R-tool?
13. The authors should explicitly state the novel contribution of this work and its similarities and differences with their previous publications.
14. I suggest the authors add a 'Conclusion' paragraph at the end of the paper. In addition, the authors need to articulate clearly the implications of the research results for theory and practice in the article. A detailed explanation of the author's recommendations should be included. I would suggest writing a paragraph in the conclusion section for the implications.
15. The authors need to clearly articulate the limitations, and Future research directions should be proposed.
16. For readers to quickly catch your contributions, it would be better to highlight major difficulties and challenges and your original achievements to overcome them in a clearer way in the abstract and introduction.
17. How could/should your study help future studies?
If these revisions can be made to the manuscript, I believe that this study can be accepted for publication.
I wish the authors all the very best with this study.
Author Response
REVIEWER 1
After the review process, the reviewer would like to give some critical thinking and idea to help authors get their job done efficiently.
We are very grateful for the time spent to carefully read our article. Thank you.
I have only a few concerns about the paper and some suggestions that maybe the authors could consider:
- To begin with, there are some typos and grammar mistakes. Some long sentences could make readers confused.
We are surprised by this comment. As English is not our first language the article has been check for English by a professional before the submission. The authors provided the bill to the editing service.
- In the 'Introduction' section, the proposed research gap and the stated objectives do not meet the criteria of proper synergy. Please make the research gap and the research objectives consistent with each other.
You are right; there could be a lack of synergy between the objectives of the paper and the proposed research gap. We brought precisions concerning the objective of the article by adding the following sentence (lines 54-58 of the revised manuscript) « […] we wonder how the question of forest insurance has been tackled in the literature by looking at (i) the methodologies mobilized, (ii) the issues addressed, (iii) the risks considered, (iv) the countries where the question is dealt with, and (v) the keywords provided by the authors. » The research gaps are a result of the literature review and consequently, they are discussed in Section 4 dedicated to discussion. However, we have now provided evidence in the introduction on this point.
- In the 'Introduction' section, I would like to suggest the authors add a paragraph at the end of the 'introduction' to write the organization of the paper to ease readers to catch the overview of the article's content. You may check an example in this ref. 'current trends in the application of eeg in neuromarketing: a bibliometric analysis'.
We added such a paragraph at the end of the introduction.
- I think the 'Material and Methods' section can be improved by adding a new bibliometric analysis of articles from different fields to highlight the importance of bibliometric analysis, which has recently become widely used, which can be beneficial for this issue.
Thank you for the suggestion. We added the footnote 2 on that point: « Such systematic search has already been conducted in economics to tackle problematics that are close to our, see for example [9-12]. »
With:
[9] Montagné-Huck, C.; Brunette, M. Economic analysis of natural forest disturbances: a century of research. J For Econ 2018, 32, 42-71. DOI : 10.1016/j.jfe.2018.03.002
[10] Brunette, M.; Bourke, R.; Hanewinkel, M.; Yousefpour R. Adaptation to climate change in forestry: a multiple correspondence analysis (MCA). Forests 2018, 9(1), 20-34. DOI: 10.3390/f9010020
[11] Iyer, P.; Bozzola, M.; Hirsch, S.; Meraner, M.; Finger, R. Measuring farmer risk preferences in Europe: a systematic review. J Agr Econ 2020, 71(1), 3-26. DOI: 10.1111/1477-9552.12325
[12] Bastit, F.; Brunette, M.; Montagné-Huck, C. Pests, wind and fire : A multi-hazard risk review for natural disturbances in forests. Ecol Econ 2023. DOI: 10.1016/j.ecolecon.2022.107702.
- It might be appropriate for the authors to explain why they had chosen the period of extracting data between 1928 and 2021.
Knowing the limited literature on the subject, we initially decided not to impose a constraint on the targeted publication period. This assumption was confirmed by our research and that is why we scanned a very long period. This period is not a choice, it is a result of our research. The search was terminated in June 2022 and the result was that we collected 38 articles with the first article published in 1928 and the last in 2021. We modified the sentence accordingly in the revised version of the paper (lines 71-73).
- Compared with the period, I think the number of extracted articles (i.e., 38) is quite small. So, the authors should clarify that.
Our bibliometric study focuses on a very specific topic on which, as specialists, we had a preconception of a very limited literature. We knew that the literature was narrow on the topic and, we expected that the number of articles was not so important. Consequently, collecting 38 articles is not surprising.
An additional reason may be that we collected only articles published in peer-reviewed journal in English. All the other documents were not included in our analysis, and would increase for sure the number of collected documents.
- Why have you chosen these specific keywords, i.e., "Insurance AND Forest". I think it's broad keywords.
Our main objective is to realize a literature review on the topic of forest insurance, so that the two keywords emerge spontaneously. We do not identify more relevant keywords than these two ones. We added a sentence on that point in the conclusion when tackling the limitations of the study.
- Figure 1 needs more clarity for public readers. For example, X-axis and Y-axis refer to what?
We have provided more details in Figure 2 (in the revised version). X-axis represents the years and Y-axis the number of publications. We added the title of the axis on the figure.
- The authors did not refer to the type of extracted documents that have been analyzed (e.g., article, review paper, ..., etc.). In addition, the authors should refer to the language of documents that have been analyzed in this study. So, I suggest a ref. 'neuromarketing research in the last five years- a bibliometric analysis', which can benefit and improve the issue.
These information are indicated in the article, lines 74-75 (in the revised version): « we only selected articles published in a peer-reviewed journal in English.».
As suggested, we also added the following figure which explains our process of selecting articles for our bibliometric analysis.
Stage |
Selection |
Exclusion |
Identification |
Records identified through Google Scholar searching |
Query: « Insurance » and « forest » |
|
|
|
Screening |
Record screened based on title, keywords and year |
Excluded due to non-compliance with the area |
|
|
|
|
Record screened based on type of publication |
Excluded not only peer-reviewed articles |
|
|
|
Eligibility |
Full articles published in English |
Excluded due to non English language |
|
|
|
Selection |
Articles included in the bibliometric study |
|
Figure 1. Flow chart to identify articles for the bibliometrics study.
- Could the authors clarify in the article what type of documents has been excluded from the Google Scholar database for this study?
As indicated in the previous answer, we added the figure to explain our article selection process. We only collected articles published in peer-reviewed journals in English. All the other documents (working paper, report, etc.) were excluded. We added a sentence in the conclusion on that point concerning the limitations of our study.
- The development of search criteria does not justify why the decisions are made. So, the authors should clarify why they have chosen the Google Scholar database for extraction papers and not in Scopus or WoS database.
We agree that this is a limitation of our study. However, we opted for this research tool because we already had a precise knowledge of the existing literature on the subject and we had strict selection criteria more in line with such a tool. We added a sentence in the conclusion on that point concerning the limitations of our study, and a footnote 1.
- Could the authors explain why they have used VOSviewer software rather than R-tool?
We already have used VOSviewer for another literature review, so that we are more familiar with this software. However, the same analysis may be conducted with R-tool.
- The authors should explicitly state the novel contribution of this work and its similarities and differences with their previous publications.
This article is original in the sense that it is the first time that a literature review is realized on the topic of forest insurance. We already work on this topic for several years but the synthesis of the literature was never realized. This special issue of Forests seems to us an opportunity to conduct and publish such a study.
- I suggest the authors add a 'Conclusion' paragraph at the end of the paper. In addition, the authors need to articulate clearly the implications of the research results for theory and practice in the article. A detailed explanation of the author's recommendations should be included. I would suggest writing a paragraph in the conclusion section for the implications.
We added a conclusion in the revised version.
- The authors need to clearly articulate the limitations, and Future research directions should be proposed.
This is now in the conclusion.
- For readers to quickly catch your contributions, it would be better to highlight major difficulties and challenges and your original achievements to overcome them in a clearer way in the abstract and introduction.
Indeed, as suggested, we added significant and fundamental elements of our analysis in both the abstract and the introduction, including the identification of gaps in the literature and the discussion of several avenues for future research.
- How could/should your study help future studies?
This literature review is a starting point. This is a necessary first step before pursuing research on forest insurance. This synthesis being done, new questions should be addressed such as the reasons of the difference in terms of insurance adoption between countries and the way the current insurance scheme should be adapted to face the new challenge of climate change.
We added some sentences on that point in the conclusion.
If these revisions can be made to the manuscript, I believe that this study can be accepted for publication. I wish the authors all the very best with this study.
Thank you.
Best regards,
The authors.

Reviewer 2 Report
Abstract
Line 8: Please rewrite the aim of the paper. The authors should revise the abstract to improve the language.
Introduction
Line 21: Place the in-cite citation at the end of the sentence. It may look better as this is the early paragraph of the section.
Line 24: Replace "computed" with an appropriate word.
Line 36-38: Missing citations. Authors should add sources for these facts.
Line 41-44: Rewriting the sentence is a must, particularly the organisation name and year.
Line 50-56: Authors should specify forest insurance's importance using more literature. I recommend the authors spell out the research aim and objectives.
Both of these are missing.
Materials and Methods
- Please mention the process of article selection.
- A systematic flow chart of article selection from title, abstract and final screening will benefit future researchers or readers.
Results
- Please reorganise the result section according to the aim and objectives of your research.
- A proper title of the sub-heading: "3.1. The methods mobilized" is a must required. I suggest authors specify the objectives in the beginning of their paper to guide the result presentation. This section should be rewritten to summarise the key methods applied so far. A paragraph and a summary table may suffice this.
- Likewise, "3.4.The countries"- not an appropriate heading
- Please follow my suggestion for the rest of the result sections with a proper sub-heading or title of the section
- Line 160 and 171, the authors used two tenses to write "observed" and "observe". Please be consistent with the appropriate tense throughout the manuscript.
Discussions
- The current discussion does not follow specific objectives or reflect the importance of results and arguments for future needs.
- The authors should mention the critical knowledge available and gaps within the discussion.
Conclusion
- Please add a conclusion section by extending the recommendations on what should be done and future directions.
Author Response
REVIEWER 2
Thank you very much for the suggestions and comments about our paper. It allows to improve its quality.
Abstract
Line 8: Please rewrite the aim of the paper. The authors should revise the abstract to improve the language.
We simplified the line 8. However, as English is not our first language, the article has been checked for English by a professional before the submission. The authors provided the bill to the editing service.
Introduction
Line 21: Place the in-cite citation at the end of the sentence. It may look better as this is the early paragraph of the section.
Done.
Line 24: Replace "computed" with an appropriate word.
The word « computed » was replaced by « indicated ».
Line 36-38: Missing citations. Authors should add sources for these facts.
As required, we added references.
Line 41-44: Rewriting the sentence is a must, particularly the organisation name and year.
The sentence was reformulated.
Line 50-56: Authors should specify forest insurance's importance using more literature. I recommend the authors spell out the research aim and objectives.
Both of these are missing.
The literature on the topic of forest insurance is very narrow, as shown in our literature review, only 38 articles published in peer-reviewed journals in English between 1928 and 2021. Using more literature in the introduction to justify the importance of forest insurance means using most of the articles included in the literature review. We prefered to keep them to be synthetized in the core of the article.
The objective is to have an overview of the current situation in the literature concerning forest insurance (lines 53-54 of the revised version). We precised the five sub-questions addressed in the article by adding this paragraph: « In particular, we wonder how the question of forest insurance has been tackled in the literature by looking at (i) the methodologies mobilized, (ii) the issues addressed, (iii) the risks considered, (iv) the countries where the question is dealt with, and (v) the keywords provided by the authors. » (lines 54-58).
Materials and Methods
- Please mention the process of article selection.
As indicated in Section 2 (lines 74-75), we considered all the articles published in peer-reviewed journal in English. All the other documents are excluded. We make this paragraph more explicit. In addition, in order to be more precise about our article selection process, we added a figure explaining our approach (Fig. 1).
- A systematic flow chart of article selection from title, abstract and final screening will benefit future researchers or readers.
Indeed, we added in the Materials and Methods section the following figure explaining our selection.
Stage |
Selection |
Exclusion |
Identification |
Records identified through Google Scholar searching |
Query: « Insurance » and « forest » |
|
|
|
Screening |
Record screened based on title, keywords and year |
Excluded due to non-compliance with the area |
|
|
|
|
Record screened based on type of publication |
Excluded not only peer-reviewed articles |
|
|
|
Eligibility |
Full articles published in English |
Excluded due to non English language |
|
|
|
Selection |
Articles included in the bibliometric study |
|
Figure. Flow chart to identify articles for the bibliometrics study.
Results
- Please reorganise the result section according to the aim and objectives of your research.
We followed your suggestion. We made explicit the link between the sub-objectives in the introduction (from (i) to (v)) and the five sub-sections of the results section.
- A proper title of the sub-heading: "3.1. The methods mobilized" is a must required. I suggest authors specify the objectives in the beginning of their paper to guide the result presentation. This section should be rewritten to summarize the key methods applied so far. A paragraph and a summary table may suffice this.
We followed your recommendation. We modified the title of the sub-heading. We replaced the text by a table (Table 2 in the revised version of the manuscript) summarizing the methods, objectives and the reference of the articles for each methodology: theoretical models and empirical analysis.
- Likewise, "3.4. The countries"- not an appropriate heading.
We modified the title of the sub-heading.
- Please follow my suggestion for the rest of the result sections with a proper sub-heading or title of the section
We provided proper titles for all the sub-headings of the results section.
- Line 160 and 171, the authors used two tenses to write "observed" and "observe". Please be consistent with the appropriate tense throughout the manuscript.
The English was corrected by a professional. The first « observe » is at present tense because « we currently observe » whereas the second « observed » is at past tense because « the authors observed » at the time of the writing of the article in 2014.
Discussions
- The current discussion does not follow specific objectives or reflect the importance of results and arguments for future needs.
We have modified the discussion section to be clearer and to specify the objectives of the discussion. We removed the first paragraph of the discussion to put it in the conclusion section. Indeed, two reviewers among three required a conclusion with a summary of the results, the limitations of the study and future research. Consequently, we added a last section dedicated to the conclusion. We also defined “front of science” as required by Reviewer 1.
- The authors should mention the critical knowledge available and gaps within the discussion.
We added a short paragraph in the conclusion about the limitations of our study. The research gaps are part of the discussion section.
Conclusion
- Please add a conclusion section by extending the recommendations on what should be done and future directions.
We wrote a conclusion section.

Reviewer 3 Report
Please explain why only the Google Scholar database has been used, and not any other complementary databases such as Scopus? Why are only two keywords used - Insurance and forest - and other synonyms, such as timberlands, are excluded?
Table 1. Listing many journals one by one from the fourth row onwards did not give me any added value, why is it necessary? On the basis of which logic is the list of journals presented in Table 1?
Results:
Why is subchapter 3.1. only presented as a list of bullets?
Is the bullet row 89 at the same level as row 88? Do all bullets have to be at the same level?
Something is missing under the bullets in lines 87 and 98.
Row 164 "pine beetles". I have been taught, and some forestry journals insists on it, that when the name of a species is given, the corresponding name in Latin must also be given. Why is that in the article ‘Pine beetles’ is mentioned by name, when in many European countries one of the most important forest pests is the European spruce bark beetle (Ips typographus)?
What is the logic behind the list of countries in the first paragraph of section 3.4?
Table 2 is in fact a repetition of the previous paragraph, it does not provide any new or additional information. Rather, I would suggest that the authors change the wording of the preceding paragraph so that the continents are included.
Line 179. What is the difference between "Sum is equal to 31" vs "sum is 31"?
Figure 2. Please explain unambiguously the meaning of the colors in this figure. "Furthermore, the color represents evolution over time, from blue in 2005 to yellow in 2020." "Themes related to forest resources and European countries are in blue, while recent appearances in yellow are about ecosystem services and resilience"
Discussion.
Overall, the discussion section is well written, but some questions were raised.
The authors have used the term "front of science". The word "front" has many meanings, some of which are related to the military matters, for example. Please clarify the meaning of this expression and why it is given in that wording in current manuscript. What is the difference between 'front of science' vs 'other ambitions and problems' (row 327)?
Why are only the French examples mentioned by name (rows 264, 303 and 333)? As a reader, I would expect examples from different regions, e.g., Eastern European countries (Slovakia, etc.)? If the article focuses only on examples from France, either the title, the keywords and/or abstract has to be changed.
In my opinion, the discussion can be better written linguistically. E.g., ‘In some countries, such as France, where 75% of forest land is private forest, it may be beneficial to financially compensate owners for their carbon sequestration efforts as part of forest management.’ Where is the focus of the previous statement? Would the above statement also apply to situations where the share of private forestry accounts for 50% or 30%?
Regarding the last paragraph of the discussion, please explain in more detail what is meant in terms of carbon loss and its compensation in the context of forest management. For example, clear-cutting is a part of forest management, the roundwood assortments or carbon that is produced, is removed from the forest – is there still a need to compensate for this financially through insurance payments?
Author Response
REVIEWER 3
Your comments and suggestions are very relevant for us and allows to improve the article. We are very grateful for that.
Please explain why only the Google Scholar database has been used, and not any other complementary databases such as Scopus? Why are only two keywords used - Insurance and forest - and other synonyms, such as timberlands, are excluded?
We opted for the Google Scholar research tool because, first, our bibliometric study focuses on a very specific topic on which, as specialists, we had a preconception of a very limited literature and, second, we already had a precise knowledge of the existing literature on the subject and we had strict selection criteria more in line with such a tool.
However, this is clearly a limitation of our study, a voluntary limitation. We added this limitation in a short paragraph in the conclusion section. We also deal with the limitation concerning the two keywords used.
Our main objective is to realize a literature review on the topic of forest insurance, so that the two keywords emerge spontaneously. We did not identify more relevant keywords than these two ones. « timberland » may be possible, however associated to insurance (the main topic of the article), we think that the same articles would appear.
Table 1. Listing many journals one by one from the fourth row onwards did not give me any added value, why is it necessary? On the basis of which logic is the list of journals presented in Table 1?
The idea was to show the variety of journals used to publish articles on forest insurance. We agree that it was clumsy. We proposed to reduce Table 1 to the first three rows and to give some examples of other journals in the text (lines 91-96). We hope that these modifications fit your expectations.
Results
Why is subchapter 3.1. only presented as a list of bullets?
This sub-chapter is now mainly represented as a table, as required by Reviewer 2.
Is the bullet row 89 at the same level as row 88? Do all bullets have to be at the same level?
The presentation as a table allows to clarify this point.
Something is missing under the bullets in lines 87 and 98.
The presentation as a table allows to clarify this point.
Row 164 "pine beetles". I have been taught, and some forestry journals insists on it, that when the name of a species is given, the corresponding name in Latin must also be given. Why is that in the article ‘Pine beetles’ is mentioned by name, when in many European countries one of the most important forest pests is the European spruce bark beetle (Ips typographus)?
We agree with you on the most important pests, but we have only quoted « pine beetles » in reference to the following paper (article [25]) where it is mentioned like this in the title:
Chen, X.; Goodwin, B.; Prestemon, J. How to fight against southern pine beetle epidemics: An insurance approach. Can J Agr Econ 2019, 67(2), 196-213. DOI: 10.1111/cjag.12186
What is the logic behind the list of countries in the first paragraph of section 3.4?
This section deals with the countries where case studies for forest insurance are considered. The idea was then to mention all the countries where the topic was addressed. This information is then aggregated at the continental level to build the corresponding table (Table 2 in the first version). We followed your suggestion and remove the table. The information are thus directly in the text (lines 180-181).
Table 2 is in fact a repetition of the previous paragraph, it does not provide any new or additional information. Rather, I would suggest that the authors change the wording of the preceding paragraph so that the continents are included.
We followed your suggestion and removed Table 2 from the text.
Line 179. What is the difference between "Sum is equal to 31" vs "sum is 31"?
We corrected.
Figure 2. Please explain unambiguously the meaning of the colors in this figure. "Furthermore, the color represents evolution over time, from blue in 2005 to yellow in 2020." "Themes related to forest resources and European countries are in blue, while recent appearances in yellow are about ecosystem services and resilience".
We added precisions in this paragraph to increase understanding.
Discussion
Overall, the discussion section is well written, but some questions were raised.
The authors have used the term "front of science". The word "front" has many meanings, some of which are related to the military matters, for example. Please clarify the meaning of this expression and why it is given in that wording in current manuscript. What is the difference between 'front of science' vs 'other ambitions and problems' (row 327)?
« Front of science » are for us topics that are increasing in economics and for which more attention should be paid in a near future. We precised this point in the introduction of the discussion (lines 216-217).
However, near these fronts of science, some topics are also interested but more secondary, explaining why we separate the « front of science » and the « other ambitions and problems ».
Why are only the French examples mentioned by name (rows 264, 303 and 333)? As a reader, I would expect examples from different regions, e.g., Eastern European countries (Slovakia, etc.)? If the article focuses only on examples from France, either the title, the keywords and/or abstract has to be changed.
The article is of course more general than France. Sorry for this clumsiness.
In line 264, we cite an original article applying index insurance to forest. To our knowledge, this is the only one on this topic, so that it will be difficult to cite other ones.
In line 303, we remove the example from France, since the case is quite similar in most of European countries.
In line 333, we cite France because this country is an exception. Indeed, the private forest area covers 75% of the territory and the public forest area is then a minority. In most of the other European countries, the opposite occurs. As a consequence, the compensation issue is highlighted in France. We change the sentence to avoid citing France.
In my opinion, the discussion can be better written linguistically. E.g., ‘In some countries, such as France, where 75% of forest land is private forest, it may be beneficial to financially compensate owners for their carbon sequestration efforts as part of forest management.’ Where is the focus of the previous statement? Would the above statement also apply to situations where the share of private forestry accounts for 50% or 30%?
What is important in this statement is the idea that when the private forest area is important in a country (whatever the percentage), the sequestration bears on private forest owners, and then, to encourage them to consider carbon sequestration in their management, it may be necessary to pay them. Of course, the greater the proportion of the private area, the greater the issue.
We rewrote the sentence.
Regarding the last paragraph of the discussion, please explain in more detail what is meant in terms of carbon loss and its compensation in the context of forest management. For example, clear-cutting is a part of forest management, the roundwood assortments or carbon that is produced, is removed from the forest – is there still a need to compensate for this financially through insurance payments?
The sentence was clumsy.
Following Song and Peng (2019), it is the forestry carbon sink that may be secure through insurance contract. The idea is the following one: forests are exposed to natural events affecting carbon sequestration in forests and reduces the output of forestry carbon sinks. In that case, forest insurance can « ensure the rapid recovery of forestry production and operation after the disaster, reduce the risk of forestry investment and financing, and promote the sustainable operation of forestry » (Section 1, Song and Peng, 2019).
Manley and Watt (2009) tackle the problematic through the case study of New-Zealand. In this country, the forest insurance contracts specify a sum insured per hectare. The idea that is being considered for the inclusion of carbon is to increase the sum insured per hectare to cover the carbon value as well as the tree crop value.
We partly reformulated the paragraph.

Round 2
Reviewer 1 Report
After this revision, I have only a few concerns about the paper and some suggestions that maybe the authors could consider:
1. I think the 'Material and Methods' section can be improved by bibliometric analysis paper from different fields to highlight the importance of bibliometric analysis, which has recently become widely used. I suggest some references of bibliometric analysis for example, neuromarketing field 'neuromarketing research in the last five years- a bibliometric analysis' and healthcare field 'nine years of mobile healthcare research: a bibliometric analysis', which can be beneficial for this issue.
2. It might be appropriate for the authors to explain why they did not include the leading academic institution in the forest and insurance area.
3. It might be appropriate for the authors to explain why the authors did not highlight the leading authors in the related area.
4. Figure 1 has some text that is hard to read; I suggest the authors to re-design figure 1 to be readable [the authors use dark background which make the font is not clear].
5. There are many bibliometric factors missed in the article such as the citation analysis, co-citation analysis, and co-citation network and clusters. The authors should briefly explain why these factors are missed or not included in the study?
6. The authors used VOSviewer software in the bibliometric analysis, however the methodology for this is not clearly explained in the Materials and Methods section. The authors should explain what is this VOSviewer and how/why did they used it, so that the readers can easily understand it.
7. Table 2 is inserted as a picture/screenshot - This should be changed and formatted as a table instead.
If these revisions can be made to the manuscript, I believe that this study can be proposed for further consideration in the journal.
I wish the authors all the very best with this study.
Author Response
Answers to Review 1 – Round 2
Thank you again for the time spent to read our article. We consider all your comments in order to improve the quality of our paper.
We answer point by point to your comments.
After this revision, I have only a few concerns about the paper and some suggestions that maybe the authors could consider:
- I think the 'Material and Methods' section can be improved by bibliometric analysis paper from different fields to highlight the importance of bibliometric analysis, which has recently become widely used. I suggest some references of bibliometric analysis for example, neuromarketing field 'neuromarketing research in the last five years- a bibliometric analysis' and healthcare field 'nine years of mobile healthcare research: a bibliometric analysis', which can be beneficial for this issue.
We thank you for this remark; indeed, bibliometric analyses are increasingly useful and used to gain insight into the literature on a specific topic. We have used the two papers indicated to improve our materials and methods section.
- It might be appropriate for the authors to explain why they did not include the leading academic institution in the forest and insurance area.
As suggested, we completed the bibliometric analysis by focusing on authors. In our analysis there are few authors and therefore few institutions, so we opted to add only an analysis by author and not by institution, as the latter is strongly correlated with authors and is too redundant. Using your comment, we added Figure 2 in the revised version of the manuscript. The figure represents the networks in terms of authors and co-citations. The networks are clearly small, gathering few authors. In that context, it seems to us that indicating the leading institution doesn’t make sense. We prefer to focus on the countries where the case studies take place.
- It might be appropriate for the authors to explain why the authors did not highlight the leading authors in the related area.
As previously indicated, we added Figure 2 in the revised version of the manuscript. The figure represents the networks in terms of authors and co-citations.
- Figure 1 has some text that is hard to read; I suggest the authors to re-design figure 1 to be readable [the authors use dark background which make the font is not clear].
We agree. We change a little bit this figure.
- There are many bibliometric factors missed in the article such as the citation analysis, co-citation analysis, and co-citation network and clusters. The authors should briefly explain why these factors are missed or not included in the study?
We realized the analysis that you suggest. There are 80 different authors for the 38 articles of the data base. The networks are represented below. The bigger network, in red color, gathers 13 authors.
We add these information in Section 2 of the revised manuscript.
- The authors used VOSviewer software in the bibliometric analysis, however the methodology for this is not clearly explained in the Materials and Methods section. The authors should explain what is this VOSviewer and how/why did they used it, so that the readers can easily understand it.
VOSviewer is an open access software tool developed to construct and visualize bibliometric networks (Van Eck and Waltman, 2010). We used this software to easily and fashionably visualize the links (in terms of publications) between the authors (Fig. 2) and the keywords (Fig. 3).
We added this sentence in footnote 3.
- Table 2 is inserted as a picture/screenshot - This should be changed and formatted as a table instead.
We corrected in the revised version of the manuscript.
If these revisions can be made to the manuscript, I believe that this study can be proposed for further consideration in the journal.
I wish the authors all the very best with this study.

Reviewer 2 Report
-Please revise the lines 10-11 in the abstract. Line 10 should contain the context of forest insurance. Rewrite the aim of the paper with a clear goal.
-Please rewrite the line 57:
This is precisely the objective of this article.
-Please rewrite the line 58:
we wonder how the question of forest insurance.
-I don’t see any similarities between the result and discussion headings. The authors should make consistency.
-Please remove the line 415 or place earlier in the conclusion:
This article is the first step of a more general reflection around forest insurance that will pursue.
Author Response
Answers to Review 2 – Round 2
Thank you again for the time spent to read our article. We consider all your comments in order to improve the quality of our paper.
We answer point by point to your comments.
Abstract:
Abstract should be revised with a clear background statement, aim and objectives.
We rewrite the abstract as suggested. We have also, both in the abstract and in the introduction, clarified the objectives of the paper in order to be consistent with the results and discussion part.
Introduction:
Line 41:43: Please revise the citation with an appropriate reference for these lines. Citation 7 needs to be corrected.
The citation [7] is correct. In the article of Zhang and Stenger (2014), in Table 1 page 5 they clearly indicated the figures mentionned in the article : 95% of the forest covered by insurance in Sweden and approximately 40% in Finland and Norway.
However, we modifiy citation [8] since it is not a published paper. The same information is indicated in the article of Brunette, Holecy, Sedliak, Tucek, Hanewinkel (2015).
Line 54: Please revise the sentence. Not an appropriate sentence, it doesn’t make clear sense to me.
“This is precisely the objective of this article.”
We rewrite this sentence.
Line 38: Please be consistent in using “fuelwood” and “firewood”
We don’t understand this comment. In line 38, we describe insurance contract.
Line 54-58: Please revise the study aim and objectives in a complete sentence rather than fragments of sentences.
This study aimed to…
We explored the following objectives.....
We rewrite the sentences. As previously indicated, we clarified the objectives of the paper in order to be consistent with the results and discussion part.
Materials & methods:
Needs to be better organised. Language should be improved.
We follow your suggestion and provide two sub-sections. Section 2.1 is now dedicated to the methodology of the systematic search whereas Section 2.2 focuses on bibliometric information on the database. In Section 2.2, we added a part dedicated to the authors and co-citations networks (required by Reviewer 1). For more coherence, we also decided to put at the end of this section, the analysis of the keywords that we had initially at the end of Section 3.
Results and discussion:
Please present your result sections according to the research objectives. Follow the same in the discussion section.
As mentioned above, we clarified the objectives of the article by stating in the introduction as follows:
« The objectives of this study are to carry out a review of the literature on the issue of forest insurance, to produce a bibliometric overview of knowledge on this issue and thus to highlight scientific fronts. »
We have therefore specified that the results section is about the first two objectives of the paper (i.e. the literature review and the bibliometric analysis) and the discussion section deals with the last objective (a more detailed analysis of the work in order to identify seven scientific fronts). At the beginning of each section we have restated the different objectives targeted to better smooth our approach and our analysis and to facilitate the understanding of our path to achieve these different objectives.
